# Effects of N-glycan modifications on spike expression, virus infectivity, and neutralization sensitivity in ancestral compared to Omicron SARS-CoV-2 variants

**Sabrina Lusvarghi, Charles B. Stauft, Russell Vassell, Brittany Williams, Haseebullah Baha, Wei Wang, Sabari Nath Neerukonda, Tony Wang, Carol D. Weiss** *

Division of Viral Products, Office of Vaccines Research and Review, Center for Biologics Evaluation and Research, Food and Drug Administration, Silver Spring Maryland, United States of America

* carol.weiss@fda.hhs.gov

## Abstract

The SARS-CoV-2 spike glycoprotein has 22 potential N-linked glycosylation sites per monomer that are highly conserved among diverse variants, but how individual glycans affect virus entry and neutralization of Omicron variants has not been extensively characterized. Here we compared the effects of specific glycan deletions or modifications in the Omicron BA.1 and D614G spikes on spike expression, processing, and incorporation into pseudoviruses, as well as on virus infectivity and neutralization by therapeutic antibodies. We found that loss of potential glycans at spike residues N717 and N801 each conferred a loss of pseudovirus infectivity for Omicron but not for D614G or Delta variants. This decrease in infectivity correlated with decreased spike processing and incorporation into Omicron pseudoviruses. Oligomannose-enriched Omicron pseudoviruses generated in GnTI⁻ cells or in the presence of kifunensine were non-infectious, whereas D614G or Delta pseudoviruses generated under similar conditions remained infectious. Similarly, growth of live (authentic) SARS-CoV-2 in the presence of kifunensine resulted in a greater reduction of titers for the BA.1.1 variant than Delta or D614G variants relative to their respective, untreated controls. Finally, we found that loss of some N-glycans, including N343 and N234, increased the maximum percent neutralization by the class 3 S309 monoclonal antibody against D614G but not BA.1 variants, while these glycan deletions altered the neutralization potency of the class 1 COV2-2196 and Etesevimab monoclonal antibodies without affecting maximum percent neutralization. The maximum neutralization by some antibodies also varied with the glycan composition, with oligomannose-enriched pseudoviruses conferring the highest percent neutralization. These results highlight differences in the interactions between glycans and residues among SARS-CoV-2 variants that can affect spike expression, virus infectivity, and susceptibility of variants to antibody neutralization.

**Data Availability Statement:** All relevant data are within the manuscript and its Supporting Information files.

**Funding:** This project, including the salaries of all co-authors, was funded by intramural research funds from the US Food and Drug Administration. The funders had no role in study design, data collection, and analysis, decision to publish, or preparation of the manuscript.

**Competing interests:** The authors have declared that no competing interests exist.

## Author summary

The SARS-CoV-2 spike surface protein is covered in glycans that may affect its function and ability to evade antibodies. Omicron variants have over 30 mutations compared to the D614G variant, yet all 22 potential N-glycosylation sites are highly conserved. Here we compared the impact of glycan changes in the spikes of the Omicron and D614G variants on virus infectivity and neutralization. We found that loss of specific glycans in the transmembrane subunit of spike greatly reduced Omicron, but not D614G, spike expression and incorporation into pseudoviruses. Changes in the overall glycan composition also reduced the infectivity of Omicron compared to D614G in live virus and pseudovirus formats. We further show that changes in specific glycans directly or indirectly affected susceptibility of pseudoviruses to therapeutic antibodies, but the effects differed among the variants. These findings highlight differences in the interplay between glycans and spike amino acid residues among SARS-CoV-2 variants that can affect spike expression and function, as well as evasion of antibodies.

## Introduction

The spike protein on the surface of severe acute respiratory syndrome coronavirus 2 (SARS-CoV-2) mediates virus entry into host cells and is the major target for neutralizing antibodies. The trimeric spike consists of a surface subunit (S1) that contains the receptor binding domain (RBD) for the angiotensin-converting enzyme 2 (ACE2) cellular receptor and a noncovalently-associated, viral fusion subunit (S2) that catalyzes fusion between viral and host cell membranes. The spike is extensively glycosylated and has 22 potential N-linked and two potential O-linked glycosylation sites per monomer (Fig 1A and 1B) that are highly conserved among the diverse variants. Glycosylation of viral spike proteins is important for protein folding, stability, and conformational dynamics, while glycans can also shield antibody epitopes. Omicron variants have more than thirty mutations, deletions, and insertions in the spike protein relative to the Wuhan-1 variant. Elucidating the effects of glycans in different SARS-CoV-2 variants may improve our understanding of spike structure and function, which may inform strategies for vaccine design and antigen production.

The protein sequence determines the position of the N-linked glycosylation, but the glycan occupancy and composition depend on the three-dimensional protein structure and the accessibility to glycan-processing enzymes [1,2]. High-mannose glycans are minimally processed while more processing leads to hybrid and complex glycans. Generally, oligomannose glycans are prevalent when glycans are heavily clustered. Changes in the protein sequence, structure, as well as the cell substrates in which proteins are expressed, can alter the glycosylation. Mass spectrometric analysis of the different N-glycosylation sites of the Wuhan-1 and the D614G variant in different expression systems have revealed that all sites are occupied [3–10]. Most sites are predominantly complex-type glycans, whereas some sites, including N234 and N709, have been reported to be occupied by primary oligomannose glycans [3–10]. Pseudoviruses expressed in 293T cells with spikes bearing deletion of individual glycans in the Wuhan-1 and D614G strains have been previously characterized. Lower infectivity has been reported for many N-glycan deletions; however, some differences have been found among different laboratories. For example, Li et al. found a decrease in infectivity for N122Q, N343Q, N717Q, N801Q and N1074Q, whereas Huang et al. found that N122Q, N331Q, N343Q, N801Q and N1194Q in the ancestral strains had lower infectivity, with the latter two severely affected [10]. N61, N122, N234, N717, N801 and N1074 have been reported to have a high abundance of

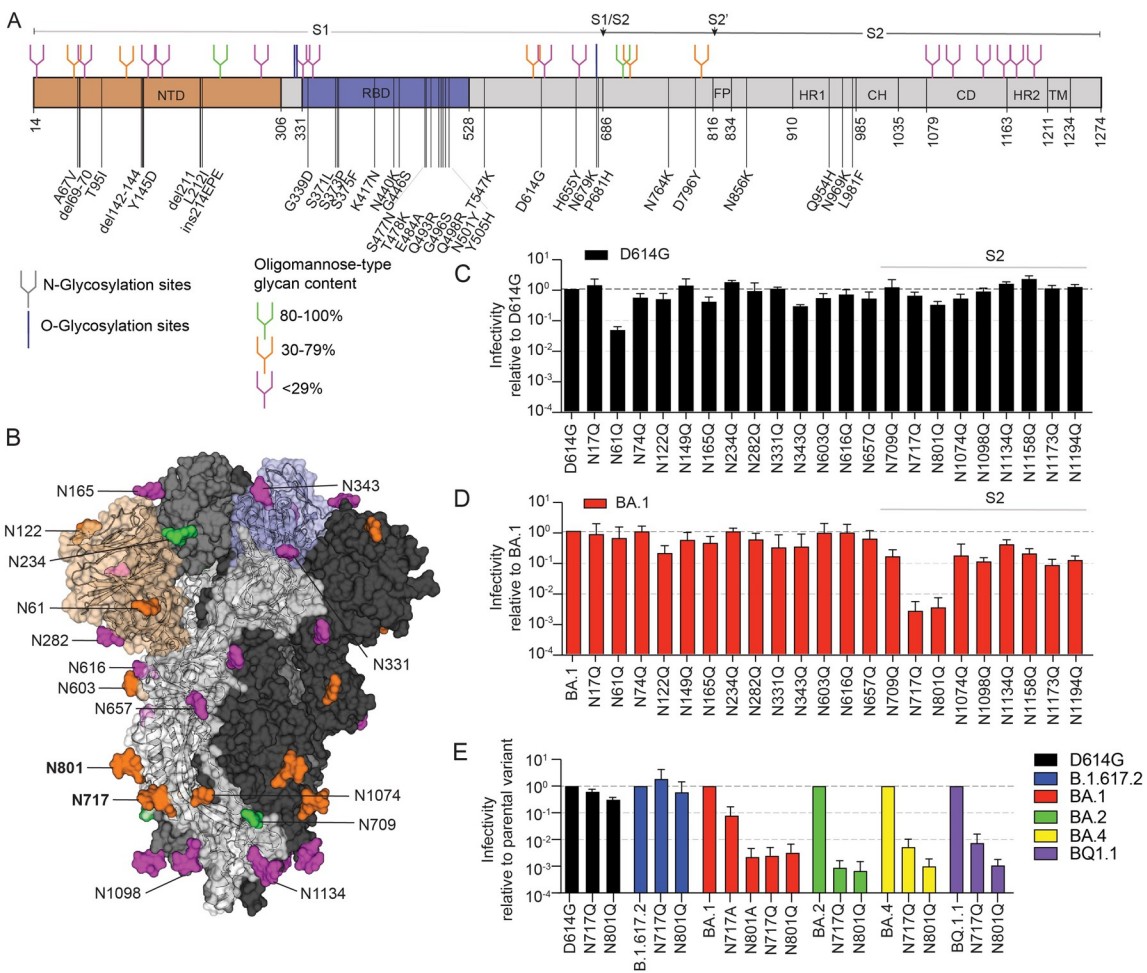

**Fig 1. Glycosylation sites in spike and effects of glycan loss on pseudoviruses infectivity.** (A) Schematic representation of the SARS-CoV-2 spike. Different spike domains are illustrated as follows: N-terminal domain (NTD, orange); receptor binding domain (RBD, blue); fusion peptide (FP), heptad repeat 1 (HR1), central helix (CH), connector domain (CD), heptad repeat 2 (HR2), and transmembrane domain (TM). Residues changed in BA.1 relative to the Wuhan-1 strain are indicated. The position of O- and N-glycosylation are shown as lines or branches, respectively. N-glycans are color-coded based on the oligomannose content at that site according to ref [4]: green, 80–100%; orange, 30–79%; and pink <29%. Furin (S1/S2) and TMPRSS2 (S2') cleavage sites are indicated by a small arrow. (B) Prefusion structure of the trimeric SARS-CoV-2 glycoprotein with all RBDs in the "down" conformation (PDB ID 6VXX). Each monomer is shown in a different shade of gray. The NTD and RBD of one monomer are shown in orange and blue, respectively. N-linked glycosylation sites are colored as described for (A) according to the percent oligomannose-type glycan content previously reported [4]. Bar graphs on panels (C) and (D) show the relative infectivity of parental and each N-to-Q substituted N-linked glycosylation site in D614G or BA.1 pseudoviruses, respectively. Infectivity was quantified by relative luciferase activity and normalized by the amount of p24 of each variant pseudovirus. Gray dotted line indicates 10-fold and 100-fold difference in infectivity. (E) The N717 and N801 residues were also changed to alanine in the BA.1 pseudovirus and to glutamine in B.1.617.2, BA.2, BA.4 and BQ.1.1 pseudoviruses. Infectivity relative to the parental spike are shown. Values represent the mean ± standard deviation (SD) of at least two independent experiments, each with an intra-assay duplicate.

unprocessed glycans, with at least 30% oligomannose glycans present at these sites [6,10]. Most of the glycosylation studies have focused on the glycosylation of the early variants, although two studies highlight differences in composition in the N- and O-glycans between the ancestral and BA.1 variants [11,12]. It remains unclear if N-glycans have divergent functional effects on the ancestral and Omicron variants.

Here we investigated how changes in glycosylation influence spike expression and the infectivity and neutralization of Omicron pseudoviruses compared to D614G pseudoviruses. We

used genetic and biochemical approaches, as well as different spike expression platforms and live SARS-CoV-2, to assess the effects of glycan modifications. Our findings highlight differences between Omicron and pre-Omicron variants in how glycans can modulate spike expression, virus infectivity, and susceptibility to neutralization by antibodies.

## Results and discussion

### Loss of N717 and N801 glycans abolish Omicron spike processing and pseudovirus infectivity

Glycans play important roles in protein folding, stability, function, and conformational dynamics. To investigate the effects of N-glycans on the function of the BA.1 spike we generated a series of pseudoviruses bearing spikes with a single asparagine to glutamine (N-to-Q) substitution that abrogated one of the 22 putative N-glycosylation sites in the spike. We also generated the corresponding spike substitutions in the D614G background as a reference and to control for glycosylation patterns that may be specific to our cell substrate and expression system. We investigated the effect of these mutations on the infectivity of D614G and BA.1 pseudoviruses in 293T-ACE2/TMPRSS2 cells (Fig 1C and 1D, respectively). We considered a 10-100-fold drop and >100 drop in infectivity relative to the corresponding unmodified variant as benchmarks for "low infectivity" and "non-infectious," respectively, as the former is well above our assay precision and the latter is near the assay background [13]. Overall, D614G pseudoviruses with N-to-Q substitutions showed less than 10-fold drop in infectivity, except for N61Q, which conferred a ∼20-fold reduction in infectivity (Fig 1C). Others also reported a decrease in infectivity from removal of the N61 glycan in the context of the Wuhan-1 or D614G pseudoviruses, but the decrease was dependent on the target cell [10,14,15]. Additional mutations have also been reported to decrease the infectivity of D614G viruses, including N122Q, N343Q, N717Q, N801Q and N1074Q, but in our system these substitutions decreased infectivity less than 10-fold.

A different set of N-to-Q substitutions reduced infectivity of BA.1 pseudoviruses compared to D614G. The N717Q or N801Q substitution in the BA.1 spike each conferred a complete loss of infectivity (>100-fold drop relative to unmodified BA.1, Fig 1D). The N801 glycosylation site is near the TMPRSS2 cleavage site, and its deletion was previously shown to lead to spike degradation when introduced in the D614G background [10]. However, in our experimental conditions the infectivity of the D614G N801Q pseudovirus was only 3.3-fold lower than the parental pseudovirus. We further investigated whether the N717Q and N801Q substitutions affected the Delta (B.1.617.2), BA.2, BA.4 and BQ.1.1 spikes (Fig 1E, blue, green, yellow, and purple bars, respectively). The N717Q and N801Q substitutions in the Delta spike did not affect pseudovirus infectivity (Fig 1E, blue bars), but the corresponding changes in BA.2, BA.4 and BQ.1.1 spikes each conferred a >100-fold reduction in pseudovirus infectivity, further demonstrating that these glycans affect Omicron variants differently than the earlier variants. We also assessed BA.1 pseudoviruses with alanine instead of glutamine spike substitutions at those positions (Fig 1E, red bars). The N801A substitution conferred a similar loss of infectivity as the N801Q substitution, but the N717A substitution conferred a more modest 14-fold drop. These results indicate that the drop in infectivity from the BA.1 N801Q substitution was due to the loss of glycosylation, though the drop in infectivity due to the N717Q substitution may be due to both the loss of the glycan and the amino acid substitution.

A drop in infectivity resulting from deletion of a specific glycan can be due to (a) the spike not being properly expressed and presented on the surface of the pseudovirus particle or (b) the glycan interfering with spike function, or both. We used western blot analyses of lysates of the pseudovirus producer cells and pseudoviruses to compare the effects of the N-to-Q spike

substitutions on spike expression, processing, and incorporation into pseudoviruses for each of the N-to-Q spikes in D614G and BA.1 backgrounds. All D614G N-to-Q spikes were expressed and processed into S1 and S2 in the producer cells (Figs 2A, upper panel and S1A), though processing was least efficient for the D614G N61Q spike. Only processed spike was efficiently incorporated into the pseudoviruses (Figs 2A, middle panel and S1A). We further assessed the extent to which infectivity correlated with amount of S1 in pseudoviruses by plotting infectivity against amount of S1 in the pseudovirus particle (S1/p24). We found that the variability in S1 incorporation into pseudoviruses correlated moderately with infectivity (Fig 2B, $r^2 = 0.56$). The lower incorporation of D614G N61Q spike into pseudoviruses corresponded with the lower pseudovirus infectivity.

Western blots of pseudovirus producer cells and pseudovirus lysates with the same single N-to-Q substitutions in the BA.1 background gave different results, particularly for changes in the S2 subunit. The BA.1 N717Q and N801Q spikes showed minimal spike processing to S1 (Fig 2C, upper panel) and S2 (S1B Fig), which corresponded with the loss of pseudovirus infectivity. Analysis of glycans in positions N717 and N801 across different sarbecoviruses showed that these two sites have high variability in the glycan composition [16]. Removal of glycosylation sites in S2 has been previously shown to lead to the expression of misfolded protein that is stacked in the ER [17]. It is possible that the BA.1 N717Q and N801Q spikes are misfolded and unable to reach the Golgi network where both glycan and furin processing take place [18]. This might explain the failure of these spikes to reach the plasma membrane where pseudoviruses bud from the cell. We also noted that pseudovirus production was generally reduced for the BA.1 N717Q, N801Q, N1074Q and N1098Q pseudoviruses, evident by low p24 amounts when similar volumes of pseudovirus stocks were loaded onto the gels (Fig 2C, lower panel). Glycans N1098, N1134, N1158, N1173, and N1194 are in the stalk region and arranged in the post-fusion structure along the long axis with almost equal spacing [19]. They have been proposed to play an important role in controlling the dynamics of membrane fusion [20]. The variability in S1 incorporation into BA.1 pseudoviruses correlated poorly with infectivity (Fig 2D, $r^2 = 0.21$), suggesting some BA.1 N-glycans may be more important for spike function than D614G N-glycans.

The mechanisms underlying the differences in the glycans effects between pre-Omicron and Omicron variants are unclear. In high resolution structures of spike, the glycans are not fully resolved. We speculate that the large number of mutations, deletions, and insertions in the Omicron spike may alter spike folding and structure during biogenesis, which could interfere with the efficiency of glycosidases to trim and process glycans. These modifications could also affect spike stability and conformational dynamics. Comparison of the prefusion structures of the Wuhan-1 and BA.1 spikes suggests that the S2 mutations (N764K, D796Y, Q954H, N969K and L981F) do not lead to obvious structural changes [21]. In the pre-fusion conformation, glycans in residues N709, N717, N801, N1098 and N1134 form a glycan belt in the S2 region that shields the area between the S1 and the stalk (S2 Fig). The only Omicron mutation in this region is D796Y. In the post-fusion conformation, residues N709, N717 and N1098 remain nearly in the same position as in the pre-fusion conformation. However, residues D796 and N801 undergo a major shift in position in the post-fusion conformation. It is possible that the interactions between D796Y and glycan N801 affect the spike conformational changes.

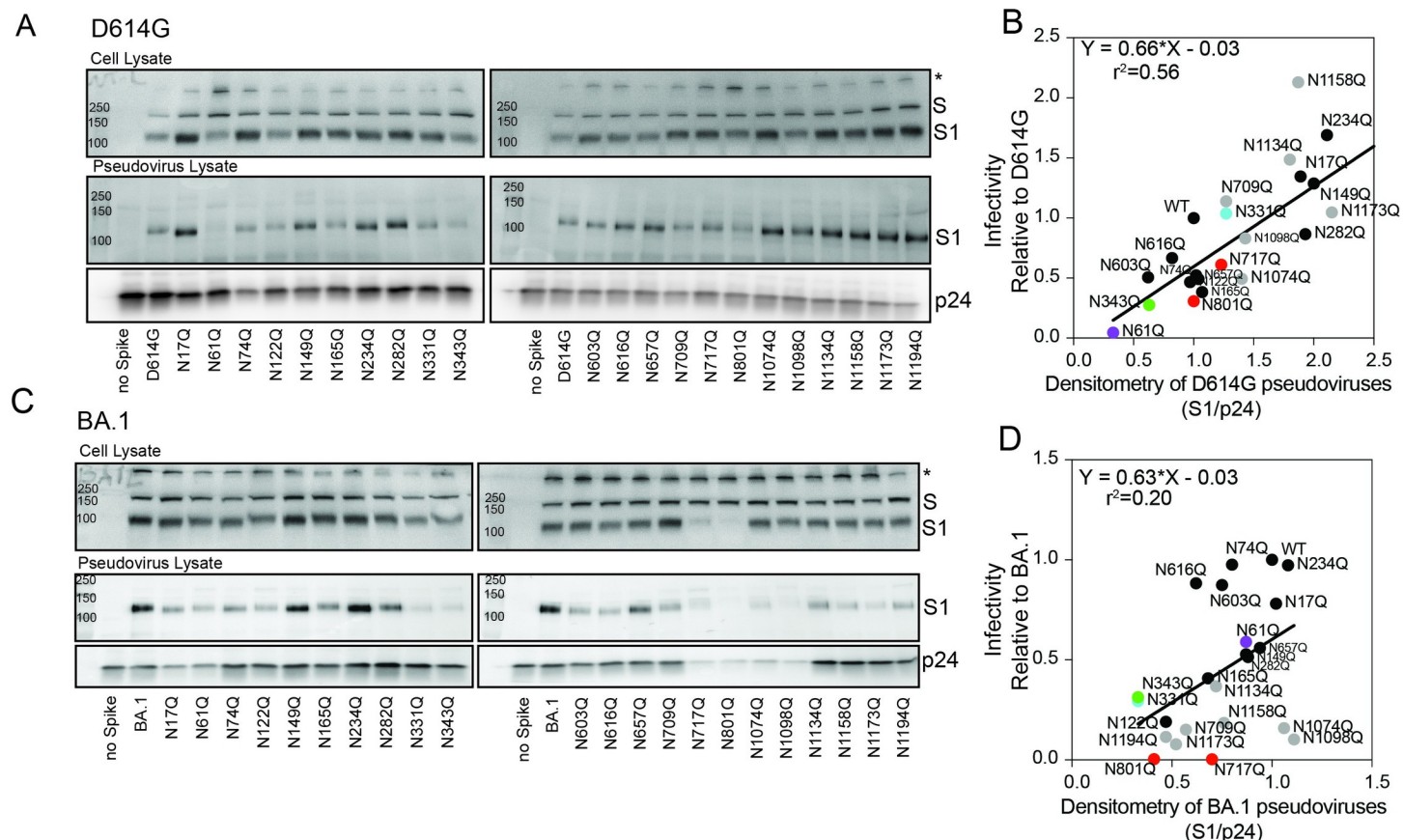

**Fig 2. Glycans N717 and N801 are essential for proper spike expression and incorporation into pseudoviruses.** Representative western blots of the (A) D614G or (C) BA.1 of the cell substrate lysates pseudovirus-producing cells (top) and pseudoviruses (bottom) are shown. Equal volumes of pseudovirus stocks were loaded on gels. The spike precursor (full length, S), furin-processed spike (S1), and p24 (lentiviral capsid) bands are indicated. Asterisk (*) band likely corresponds to aggregated spikes. Anti-S1 and anti-p24 antibodies were used for detection. Correlation between the amount of S1 in each pseudovirus (S1/p24) and the corresponding infectivity for (B) D614G and (D) BA.1 are shown. The amount of S1 and infectivity were each normalized against the amount of p24 to account for differences in the amount of pseudoviruses harvested in each preparation. Densitometry values represent the average of two (D614G) or three (BA.1) independent experiments. Infectivity values represent the mean of at least two independent experiments, each with an intra-assay duplicate.

## Abrogation of complex glycosylation abolishes Omicron BA.1 and BA.2 pseudoviruses infectivity

We next investigated the effects of spike glycan modifications by generating D614, B.1.617.2, BA.1, and BA.2 pseudoviruses in 293T cells in the presence of two glycosidase inhibitors (1) kifunensine, which inhibits the ER-mannosidase I enzyme and results in $Man_9GlcNAc_2$ glycans, and (2) swainsonine, which inhibits the Golgi-$\alpha$-mannosidase II enzyme and favors truncated complex-type glycans except at glycosites that naturally contain high-mannose glycans. We also generated pseudoviruses in HEK293S GnTI⁻ cells (GnTI⁻ cells) that have been engineered to lack N-acetylglucosaminyltransferase I activity and therefore produce high-mannose glycosylation $Man_{5-9}GlcNAc_2$ that are not further modified to complex glycans. Fig 3A summarizes N-glycan processing and the inhibitors for the different enzymes.

BA.1 and BA.2 pseudoviruses produced in 293T cells had similar infectivity as D614G pseudoviruses, while B.1.617.2 pseudoviruses had 6.5-fold higher infectivity (Fig 3B). Generation of pseudoviruses in the presence of kifunensine or in GnTI⁻ cells conferred a modest loss of infectivity for D614G or B.1.617.2 pseudoviruses (Fig 3C, lanes 1 and 2). In contrast, BA.1 and BA.2

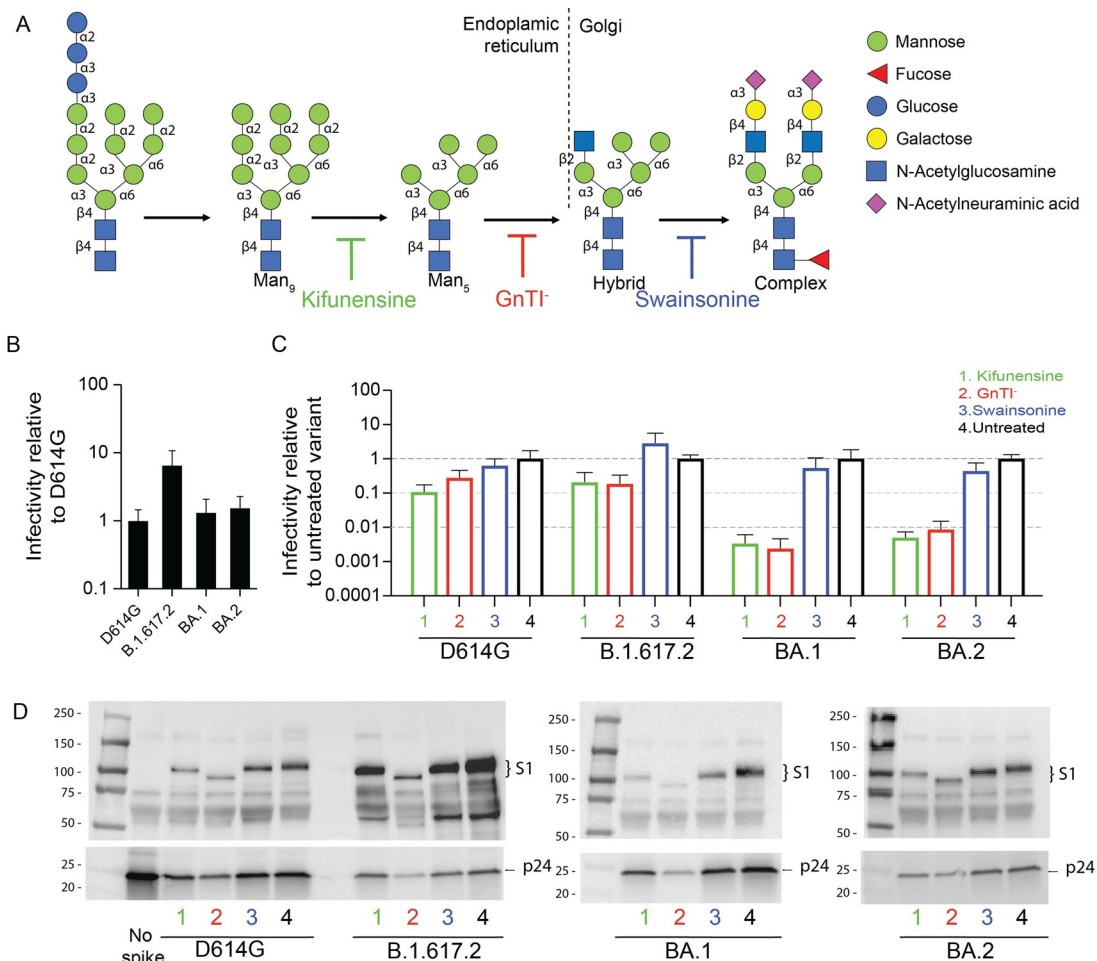

**Fig 3. Omicron pseudoviruses with oligomannose-type N-glycans are non-infectious.** (A) N-glycosylation pathway depicting the formation of high-mannose, hybrid, and complex glycans. Specific types of glycosylation can be enriched by loss or inhibition of specific glycosidases by using GnTI⁻ cells or the kifunensine or swainsonine inhibitors, respectively. (B) Infectivity of D614G, B.1.617.2, BA.1, and BA.2 variants in 293T-ACE2/TMPRSS2 cells. Infectivity was quantified by luciferase activity normalized for p24 input. The relative infectivity was calculated using D614G as reference. (C) Infectivity of D614G, B.1.617.2, BA.1, and BA.2 expressed in the presence of 50 μM kifunensine, in GnTI⁻ cells, in the presence of 20 μM swainsonine, or in untreated 293T cells. The relative infectivity was calculated using the corresponding untreated spikes as reference. Values represent the mean ± SD from at least three independent experiments. (D) Representative western blots of D614G, B.1.617.2, BA.1, and BA.2 pseudoviruses expressed in 293T cells in the presence of 50 μM kifunensine, in GnTI⁻ cells, in 293T cells in the presence of 20 μM swainsonine, or in untreated 293T cells. Anti-S1 antibody and anti-p24 antibodies were used as primary antibodies.

pseudoviruses generated in the presence of kifunensine or in GnTI⁻ cells were non-infectious, with >100-fold drops in the infectivity relative to the corresponding untreated BA.1 or BA.2 pseudoviruses (Fig 3C, lanes 1 and 2). Swainsonine did not significantly affect pseudovirus infectivity in any of the variants tested (Fig 3C, lanes 3). It has been previously suggested that spikes lacking complex glycosylation may undergo enhanced spike cleavage and S1 shedding due to proteolysis at the S1/S2 interface [22]. Additionally, it has been shown that the treatment of Wuhan-1 or D614G pseudoviruses with PNGase F, which removes most of N-linked glycosylation, significantly decreased the infectivity, whereas treatment with Endo H, which only removes high-mannose and some hybrid N-glycans, had less impact [22,23]. Collectively,

these findings indicate that complex glycosylation in one or more spike positions is important for preserving function of the BA.1 and BA.2 spikes.

Western blot analysis showed that D614G, B.1.617.2, BA.1, and BA.2 pseudoviruses bearing different glycans had processed spikes (Fig 3D). Expression of the B.1.617.2 spike was generally higher than expression of the other spike variants, regardless of the treatment. Lower amounts of p24 in the pseudovirus stocks indicated that some treatments reduced the yields of pseudovirus. As expected, spikes produced in GnTI⁻ cells had lower molecular weights than untreated spikes.

In summary, these results show that BA.1 and BA.2 spikes bearing only oligomannose glycans are poorly functional. Relative to D614G and Delta spikes, BA.1 and BA.2 spike function appears to be more sensitive to glycosylation modifications. A prior study using lectin microarrays showed that Omicron variants have a high content of galactose-containing glycans and low content of high mannose glycans, suggesting a shift towards more complex or hybrid type of structures [24]. Changes in lectin binding could enable Omicron to interact with alternative receptors such as the DC-SIGN lectin [25]. SARS-CoV-2 has been reported to use DC-SIGN though it is not known if there are differences among the variants [26,27]. Previously, the HIV antibody 2G12 was shown to bind to four high-mannose glycans at residues 709 and 801 in adjacent protomers in the Wuhan-1 spike [28], but 2G12 binding to the Omicron spike was weaker [29]. These reports align with our data suggesting that the glycan composition of Omicron spikes differs in important ways compared to spikes of earlier variants.

## Kifunensine treatment abolishes infectivity of BA.1.1, but not Delta or D614G live SARS-CoV-2

Pseudoviruses are valuable tools for assessing the effects of novel mutations in SARS-CoV-2 spikes, but they differ from live SARS-CoV-2 in ways that could affect glycosylation patterns and infectivity. Coronaviruses bud into the lumen of the ER-Golgi intermediate compartment (ERGIC) [30], whereas lentiviral pseudovirus spikes are expressed from codon-optimized genes and transported through the ER-Golgi network to the cell surface [31]. The presence of complex-type glycans in spikes of live SARS-CoV-2 suggests that the budding of coronaviruses follows an unconventional route involving linker compartments that communicate with the Golgi stacks and have access to the Golgi enzymes [30,32]. Spikes on pseudoviruses may have biosynthetic transit times, spike packing densities, and exposures to sets of glycan-modifying enzymes that differ from spikes on live SARS-CoV-2. We therefore tested the growth kinetics in Vero E6 cells of D614G, B.1.617.2, and BA.1.1 (BA.1 + R346K) live SARS-CoV-2 produced in the presence of kifunensine (Fig 4). Viruses were titered using a focus-forming assay due to poor cytopathic effects of the Omicron variant in Vero E6 cells [33]. We found that D614G and B.1.617.2 viruses grown in the presence of kifunensine had similar virus titers 2–4 days post-infection (DPI) as their untreated counterparts, but the BA.1.1 virus had >10-fold lower titers at 2 DPI and >100-fold lower at 4 DPI. These results confirm that certain glycan modifications are more important for maintaining Omicron BA.1 and BA.1.1 spike function than D614G spike function. Thus, despite having conserved glycosylation sites, the glycan requirements for Omicron variant function differ from those required for D614G and Delta variants. Our studies also highlight glycan considerations when selecting an expression system for spike structure/function studies and antigen generation.

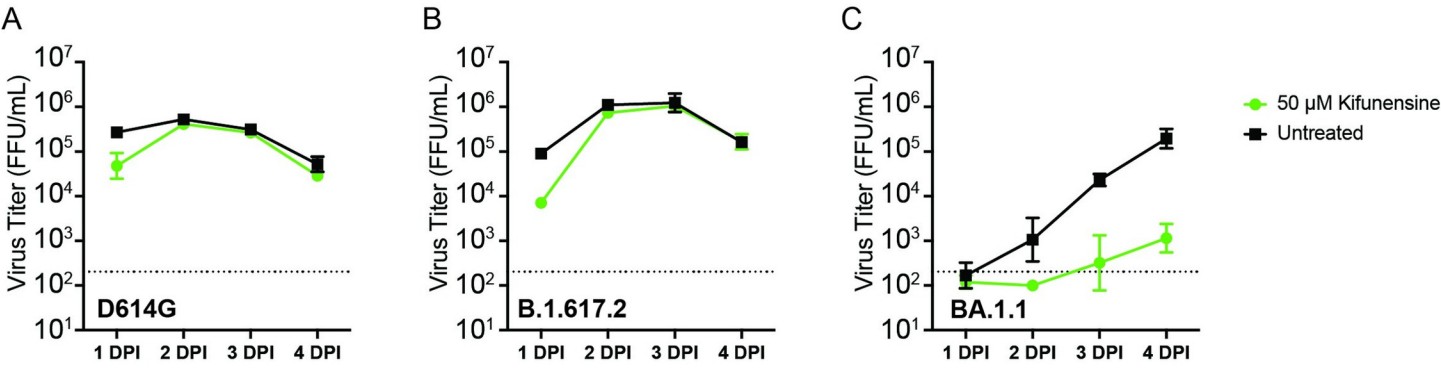

**Fig 4. Kifunensine reduces the titer of live SARS-CoV-2 BA.1.1, but not D614G or Delta variant.** D614G (A), B.1.617.2 (B), and BA.1.1 (C) live SARS-CoV-2 were produced in the absence or presence of 50 μM kifunensine. Virus titers at 2–4 days post infection (DPI) were measured as focus-forming units per mL (FFU/mL). Values represent the mean ± SD of four biological replicates.

## N-glycans deletions can alter the potency or completeness of neutralization by some therapeutic antibodies

In addition to their contributions to protein folding, stability, and function, glycans can contribute to evasion of neutralizing antibodies. We therefore assessed how changes in spike glycosylation affected the potency of five therapeutic monoclonal antibodies (mAbs), S309 (a precursor of Sotrovimab), C135 [34], Bebtelovimab [35], Etesevimab [36], and COV2-2196 (a precursor of Tixagevimab and component of Evusheld) [37] (Fig 5A and 5E). Bebtelovimab, Sotrovimab, Etesevimab (in combination with Bamlanivimab) and Evusheld were previously authorized by the US Food and Drug Administration for emergency use [38–41], but their authorizations were withdrawn due to circulating, resistant SARS-CoV-2 variants [42–44].

Previously we and others found that S309 and C135 (both Class 3 neutralizing antibodies according to the Barnes classification [34]) incompletely neutralized susceptible variants, with neutralization curves that plateaued well under 100% neutralization [27,45–47]. Partial neutralization has also been reported for some NTD antibodies [48,49]. We note that Sotrovimab has provided strong protection in humans and mouse models even though full neutralization was not achieved in some *in vitro* assays. Non-neutralizing immune mechanisms may also contribute to protection [50]. This incomplete neutralization by S309 has been attributed to target cells that express high levels of the ACE2 receptor [27,51]. However, incomplete neutralization seen for some anti-HIV antibodies has been attributed to heterogeneity in the glycosylation of the HIV envelope glycoprotein [52]. We therefore investigated if glycan heterogeneity in the spikes of the pseudovirus stocks could contribute to the incomplete neutralization by these class 3 antibodies. Because neutralization was incomplete an $IC_{50}$ could not be calculated for these antibodies. Instead, we calculated a partial $IC_{50}$ ($pIC_{50}$) based on the concentration that gave 50% neutralization of the maximal infection, even if the neutralization curve plateaued well below 100%.

Both S309 and C135 recognize RBD epitopes that contain the N343 glycan, and these antibodies do not compete with ACE2 binding to spike. Fig 5A shows a model of spike in the closed conformation with a single C135 (green), S309 (red), or Bebtelovimab (yellow) Fab bound to the RBD. Cryo-EM of the Fabs of these two antibodies bound to spike gave structures that contained one RBD up and two down. Although antibodies bound to both up and down conformations, the Fabs bound to the up RBD were weakly resolved in both cases [34,45]. Bebtelovimab has an epitope that partially overlaps the S309 and C135 epitopes [35]. In

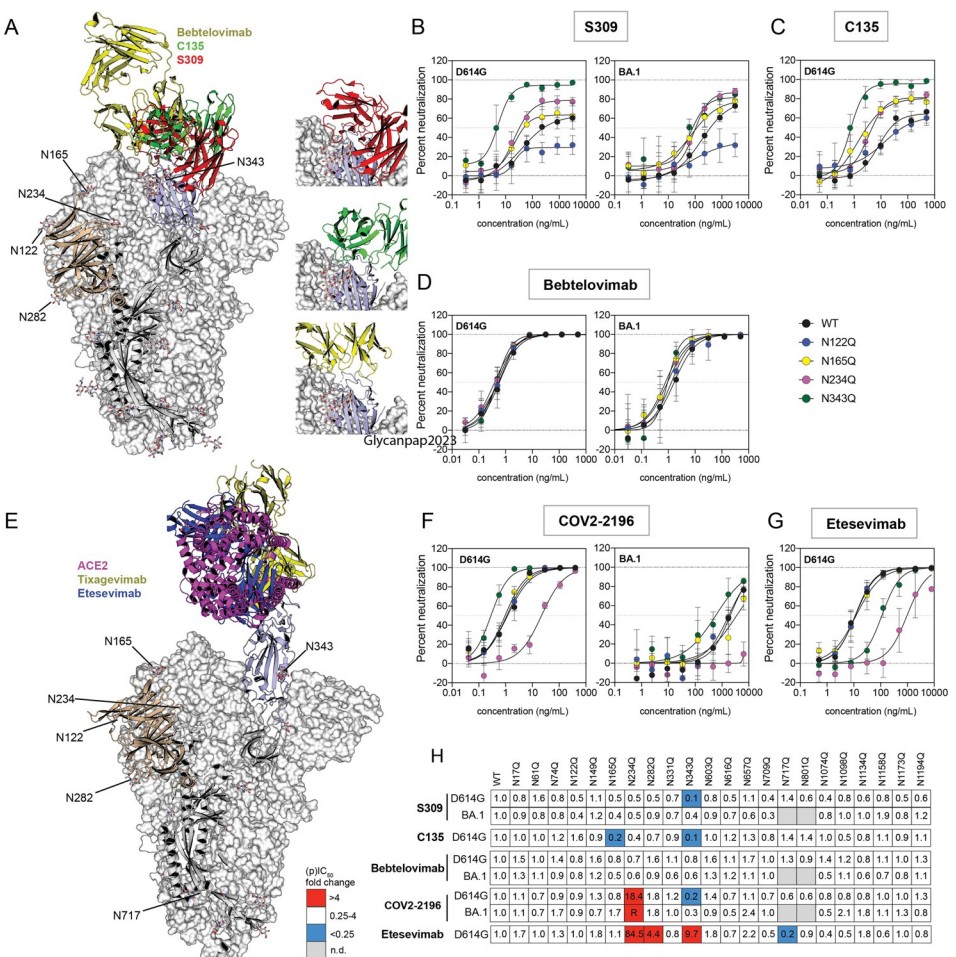

**Fig 5. Removal of specific N-glycans can alter neutralization potency and completeness.** (A) Structure of SARS-CoV-2 spike in the closed conformation complexed with Bebtelovimab (PDBID:7MM0, yellow), C135 (PDBID: 7K8Z, green) and S309 (red, PDBID: 6WPS). Neutralization curves for selected N-to-Q D614G and BA.1 pseudoviruses are shown for monoclonal antibodies (B) S309, (C) C135 and (D) Bebtelovimab. Curve colors are as follows: black, wild-type (WT)/parental D614G or BA.1 pseudoviruses; blue, N122Q; yellow, N165Q; pink, N234Q; and green N343Q substituted D614G or BA.1 pseudoviruses. (E) Structure of SARS-CoV-2 spike with one RBD in the open conformation in complex with ACE2 (PDBID:7A94, magenta), Tixagevimab (PDBID: 7L7D, yellow) and Etesevimab (blue, PDBID: 7C91). Representative neutralization curves of selected D614G and BA.1 N-to-Q pseudoviruses by (F) COV2-2196 and (G) Etesevimab. (H) Heat map showing the change in partial IC$_{50}$ (pIC$_{50}$, S309 and C135) or IC$_{50}$ (Bebtelovimab, COV2-2196 or Etesevimab) relative to D614G and BA.1. Colors as follow: red, decrease in potency >4-fold; white, no change in IC$_{50}$ (ratios between 0.25- and 4-fold) and blue increase in potency (ratios <0.25-fold). Values represent the mean ± SD from at least three independent experiments, each with an intra-assay duplicate.

contrast to S309 and C135, the Bebtelovimab epitope does not include the N343 glycan, and Bebtelovimab binding to the spike can block ACE2 binding [35].

Table 1 summarizes the maximum percent neutralization of the N-to-Q substituted D614G and BA.1 pseudoviruses by these class 3 antibodies. We found that S309 achieved only 65% and 79% maximum neutralization against the parental (WT) D614G and BA.1 pseudoviruses, respectively (Fig 5B black curves, left and right panels, respectively). C135 reached only 64% maximum neutralization against the WT D614G pseudovirus (Fig 5C, black curve) but could not be tested against the WT BA.1 pseudovirus because this variant is resistant to C135.

**Table 1. Maximum neutralization of class 3 antibodies against D614G and BA.1 N-to-Q pseudoviruses.**

| | C135 | S309 | |
|---|---|---|---|
| | D614G | D614G | BA.1 |
| **WT** | 64 ± 4 | 65 ± 7 | 79 ± 7 |
| **N17Q** | 64 ± 3 | 58 ± 2 | 74 ± 7 |
| **N61Q** | 61 ± 4 | 69 ± 7 | 73 ± 10 |
| **N74Q** | 62 ± 3 | 63 ± 6 | 77 ± 9 |
| **N122Q** | 62 ± 9 | 29 ± 4 | 36 ± 19 |
| **N149Q** | 55 ± 4 | 56 ± 5 | 83 ± 15 |
| **N165Q** | 81 ± 4 | 63 ± 3 | 79 ± 7 |
| **N234Q** | 81 ± 3 | 79 ± 3 | 87 ± 4 |
| **N282Q** | 63 ± 3 | 62 ± 3 | 75 ± 5 |
| **N331Q** | 60 ± 4 | 51 ± 3 | 68 ± 9 |
| **N343Q** | 96 ± 2 | 95 ± 3 | 82 ± 6 |
| **N603Q** | 62 ± 3 | 61 ± 3 | 77 ± 11 |
| **N616Q** | 64 ± 7 | 58 ± 5 | 74 ± 5 |
| **N657Q** | 58 ± 4 | 58 ± 7 | 75 ± 7 |
| **N709Q** | 69 ± 6 | 59 ± 7 | 75 ± 9 |
| **N717Q** | 67 ± 9 | 43 ± 6 | n.d. |
| **N801Q** | 63 ± 4 | 74 ± 7 | n.d. |
| **N1074Q** | 61 ± 4 | 45 ± 6 | 86 ± 13 |
| **N1098Q** | 69 ± 11 | 54 ± 6 | 85 ± 7 |
| **N1134Q** | 65 ± 5 | 65 ± 4 | 73 ± 15 |
| **N1158Q** | 63 ± 3 | 54 ± 4 | 86 ± 34 |
| **N1173Q** | 73 ± 14 | 50 ± 5 | 80 ± 9 |
| **N1194Q** | 66 ± 3 | 60 ± 3 | 75 ± 11 |

Deletion of the N343 glycan restored nearly complete neutralization and increased the potency for both S309 and C135 against D614G pseudoviruses but had a minimal effect on S309 neutralization of the BA.1 pseudovirus (Fig 5B and 5C, green curves). The N343 glycan, which has been reported to have mostly complex-type glycans [4], has been proposed to shield the RBD in the closed conformation and facilitate opening of the RBD [53].

A similar but more modest trend towards higher maximum neutralization by S309 and C135 was seen for D614G N234Q pseudoviruses (79% and 81%, respectively, Fig 5B left panel and 5C, pink curves) and by C135 for D614G N165Q pseudoviruses (81%, Fig 5C, yellow curve). As seen for the BA.1 N343Q pseudovirus, the maximum neutralization by S309 for BA.1 N234Q and N165Q pseudoviruses did not change (75% and 79%, respectively, Fig 5B right panel, pink and yellow curves, respectively). The N234 has been reported to be predominantly occupied by oligomannose glycans, whereas N165 and N343 glycans have been reported to be mostly complex glycans [4]. Deletion of any of these three glycans favors the RBD down conformation [6,53,54]. Molecular dynamics simulations suggest that the differences in the complex glycan-protein networks between D614G, Delta and Omicron variants can control the flexibility of the RBD domain [55]. The increase in the maximum neutralization observed for the D614G N343Q and N234Q pseudoviruses suggests that a subset of the WT D614G pseudovirus stock that resist neutralization by these antibodies may have glycan modifications at these residues that facilitate evasion of neutralization. Specific modifications of the N343 glycan could directly ease S309 and C135 binding to their epitopes. In addition, the loss of

N343 and N234, and the N165 glycan in the case of C135, may indirectly facilitate binding of these antibodies to spike by favoring the RBD down conformation.

The lack of an effect on S309 neutralization of BA.1 pseudoviruses by deletion of glycans N343, N234, and N165 suggests that the divergent spike may have a different composition of glycans or that the local structure of the BA.1 spike alters glycan conformations differently from D614G spike, or the combination. The proportion of the open and closed conformations differs between the spikes of ancestral and Omicron variants. Both the apo and the S309 bound structures of the ancestral spike with the engineered two proline substitutions (2P) have an even proportion of closed and one-up RBD structures [45], but the one-up RBD conformation dominated the apo and S309-bound 2P engineered Omicron structures. Studies with the unmodified spikes show that it can assume a wide variety of conformational states [21,29], and some Omicron substitutions seem to stabilize the closed conformation in a manner unique to Omicron [29]. Differences in the ability to bind to the ancestral spike have been shown for the S309 IgG and Fab, with the latter not reaching complete neutralization even though the potency was similar for both [45]. These differences have been attributed to trimer cross-linking, steric hindrance, or aggregation of viruses by the IgG [45]. Our data suggest that the RBD-down conformation favors the neutralization of the spike by these class 3 antibodies and that shifting the equilibrium to the open conformation may aid neutralization escape.

Receptor levels have also been implicated as a factor that can influence neutralization. Low ACE2 concentrations have been reported to favor full neutralization by NTD and class 3 antibodies [46,51]. As ACE2 binds to the spike, it shifts the conformational equilibrium towards a spike with the RBDs in the up conformation [56]. Thus, conditions that favor the closed conformation, such as removal of specific glycans or low ACE2 concentrations, may facilitate more complete neutralization by some class 3 antibodies.

While the loss of the glycan at N343 increased the maximum neutralization of S309 and C135, the loss at N122 decreased the maximum neutralization by S309 to 29% and 36% against D614G and BA.1 pseudoviruses, respectively (Fig 5B, blue curves). However, N122Q had a minimal effect on neutralization by C135 (62%, Fig 5C, blue curve). N122 has been reported to contain both processed and unprocessed glycans that can reside at the crevice between the NTD and the RBD [6]. Deletion of this glycan has been reported to support opening of spike [3,6], which likely disfavors S309 binding to spike. Deletion of this glycan has also been associated with a decrease in infectivity of the Wuhan-1 and D614G strains, though the decrease in infectivity was minimal in our conditions (Fig 1C).

In contrast to C135 and S309, Bebtelovimab was unaffected by all glycan-deleted pseudoviruses (Fig 5D). Although part of the epitope of Bebtelovimab overlaps with the S309 and C135 binding site, it engages the RBD at a different angle, so the avidity effect of the IgG might differ from those other class 3 antibodies. In addition, Bebtelovimab binds an RBD region that overlaps the receptor binding motif (RBM), which is exposed in both the open and closed conformations [35]. Collectively, these findings show that antibodies that bind to similar or overlapping regions of the RBD may be affected differently by spike glycan modifications within as well as outside of the antibody epitope and that the effects may depend on the variant context.

We further probed changes in neutralization capacity due to glycan-mediated changes in spike conformational dynamics by assessing two class 1 antibodies, COV2-2196 and Etesevimab, against our panel of N-to-Q pseudoviruses (Fig 5E, yellow and blue ribbons, respectively). Class 1 antibodies interact with the RBD in the open conformation, targeting epitopes overlapping the RBM that are occluded in the RBD down conformation [36,37,56]. We only assessed neutralization against the D614G pseudovirus for Etesevimab because this antibody lost potency against the BA.1 variant. Complete neutralization against the WT D614G pseudovirus was observed for both COV2-2196 and Etesevimab (Fig 5F left panel and 5G, black

curves, respectively). The reduced potency of COV2-2196 against WT BA.1 pseudovirus did not reach complete neutralization at the highest concentration tested (Fig 5F right panel, black curve). The potency of COV2-2196 decreased against the D614G N234 pseudovirus but increased against the D614G N343Q pseudovirus (Fig 5F left panel, pink and green curves, respectively, and Fig 5H). BA.1 N234Q pseudovirus was fully resistant to COV2-2196 in the range of concentrations tested (Fig 5F right panel, pink curve and 5H). Etesevimab lost potency against both N234Q and D614G N343Q pseudoviruses (84.5 and 9.5-fold drop, Fig 5G pink and green curves, respectively, and 5H). Our findings agree with and extend another study reporting that N234Q decreases the neutralization sensitivity to some antibodies against the Wuhan-1 variant [14]. The decreased potency of Etesevimab against the N343Q and N234Q pseudoviruses aligns with the observation that deletion of these glycans favors the RBD conformational dynamics towards the down conformation [53,54]. Substitution of N234 and N343 by alanine has also been shown to decrease ACE2 binding [53,54]. The unexpected increase in potency of COV2-2196 against the D614G N343Q pseudovirus may be a consequence of avidity differences, as this antibody could potentially crosslink adjacent RBDs as proposed for other class 1 IgGs [34].

Finally, two additional glycan deletions affected Etesevimab potency. The N282Q substitution decreased neutralization potency against D614G pseudoviruses while the N717Q substitution increased it (Fig 5H). N282 is reported to contain predominantly complex glycans, whereas N717 is reported to contain a mixture of oligomannose and complex glycans [6]. N282 is proximal to the RBD and might contribute to the shielding and dynamics of the RBD. In contrast, N717 is distant from the epitope, suggesting an allosteric effect.

## N-glycan modifications can also affect maximum neutralization by some therapeutic antibodies

Pseudoviruses produced in 293T cells in the presence of inhibitors or in GnTI⁻ cells were used to assess if glycan composition also affected neutralization. The BA.1 pseudoviruses made in GnTI⁻ cells or in the presence of kifunensine could not be assessed due to low infectivity. We found that both C135 and S309 had increased potency and reached nearly complete neutralization against D614G pseudovirus produced in GnTI⁻ cells (Table 2 and Fig 6A and 6B, red curves). Neutralization by S309, but not C135, of D614G pseudoviruses produced in the presence of kifunensine followed the same trends (Fig 6A and 6B, green curves). No changes in neutralization curves were seen against pseudovirus produced in the presence of swainsonine (Fig 6A and 6B, white curves). Earlier studies showed that some NTD antibodies showed increased potency against pseudovirus produced in the presence of swainsonine [49]. The potency of Etesevimab, COV2-2196, and Bebtelovimab was minimally affected by any of the treatments (Fig 6C and Table 2). Altogether, these data show that high-mannose glycans on spike improve neutralization potency of S309 and C135, as was seen for deletion of N165, N234, and N343 glycans. N234 has previously been shown to be mostly high-mannose [6] and is therefore unlikely responsible for the increased neutralization after treatment with kifunensine. In contrast, N343 and N165 have both been reported to have more complex glycans [6]. Thus, changing the composition of the glycans at these residues may modulate the spike conformational dynamics and affect binding by S309 and C135 but not Etesevimab, COV2-2196, and Bebtelovimab. In addition, it has been previously shown that spikes with mature glycans bind to ACE2 better than spikes with high-mannose glycosylation [22,57]. Less efficient ACE2 binding may also improve the neutralization of some antibodies.

Finally, we also assessed neutralization of SARS-CoV-2 virus-like particles (VLPs) using an expression system that uses only SARS-CoV-2 proteins [58,59]. Even though HIV-based

**Table 2. Antibody IC50 (or pIC50) and maximum neutralization of D614G and BA.1 pseudoviruses bearing different glycosylation and SARS-CoV-2 VLPs.**

| Antibody | Variant | Kifunensine | | GnTI⁻ | | Swainsonine | | Untreated | | SARS-CoV-2 VLPs | |
|---|---|---|---|---|---|---|---|---|---|---|---|
| | | pIC50 | Max % neut | pIC50 | Max % neut | pIC50 | Max % neut | pIC50 | Max % neut | pIC50 | Max % neut |
| C135 | D614G | 15.6 ± 7.8 | 55 ± 9 | 2.03 ± 0.54 | 90 ± 5 | 6.2 ± 1.7 | 59 ± 4 | 8.1 ± 1.9 | 64 ± 4 | n.d. | 19.6 ± 2.7 |
| S309 | D614G | 65.3 ± 13.1 | 76 ± 5 | 8.78 ± 1.86 | 88 ± 4 | 32 ± 16 | 58 ± 6 | 40 ± 14 | 63 ± 6 | 129 ± 188 | 31.6 ± 2.6 |
| | BA.1 | n.d. | n.d. | n.d. | n.d. | 170 ± 55 | 74 ± 7 | 135 ± 62 | 54 ± 7 | n.d. | n.d. |

| Antibody | Variant | Kifunensine | GnTI⁻ | Swainsonine | Untreated | SARS-CoV-2 VLPs |
|---|---|---|---|---|---|---|
| | | IC50 | IC50 | IC50 | IC50 | IC50 |
| Etesevimab | D614G | 11.6 ± 2.7 | 17.5 ± 3.2 | 6.89 ± 0.72 | 21.3 ± 1.9 | 43.2 ± 6.3 |
| COV2-2196 | D614G | 1.15 ± 0.28 | 0.67 ± 0.10 | 0.79 ± 0.11 | 1.18 ± 0.14 | 1.59 ± 0.22 |
| | BA.1 | n.d. | n.d. | 1654 ± 217 | 736 ± 262 | 1871 ± 327 |
| Bebtelovimab | D614G | 0.39 ± 0.08 | 0.36 ± 0.06 | 0.47 ± 0.07 | 0.55 ± 0.04 | 1.04 ± 0.13 |
| | BA.1 | n.d. | n.d. | 2.09 ± 0.15 | 2.35 ± 0.50 | 5.43 ± 0.52 |

lentiviral pseudoviruses incorporate envelope proteins at the cell surface, the HIV envelope has a high proportion of high-mannose glycans [60]. In contrast, the SARS-CoV-2 spike, which gets incorporated into viral particles during budding into the lumen of an ERGIC compartment, has a high content of complex glycans [3–8,10]. Differences in glycan compositions may be due to differences in the protein sequence, biosynthetic pathway, glycoprotein density on the viral particle, or a combination of these factors. We found that both S309 and C135 incompletely neutralized these VLPs, regardless of the variant (<30%, Fig 6D and 6E, black curves and Table 2). Bebtelovimab and COV2-2196 fully neutralized both variants (Fig 6F and Table 2). Due to the low infectivity of the VLPs (S3 Fig), we were not able to assess the VLPs made in the presence of glycosidase inhibitors. Our neutralization results suggest that the large

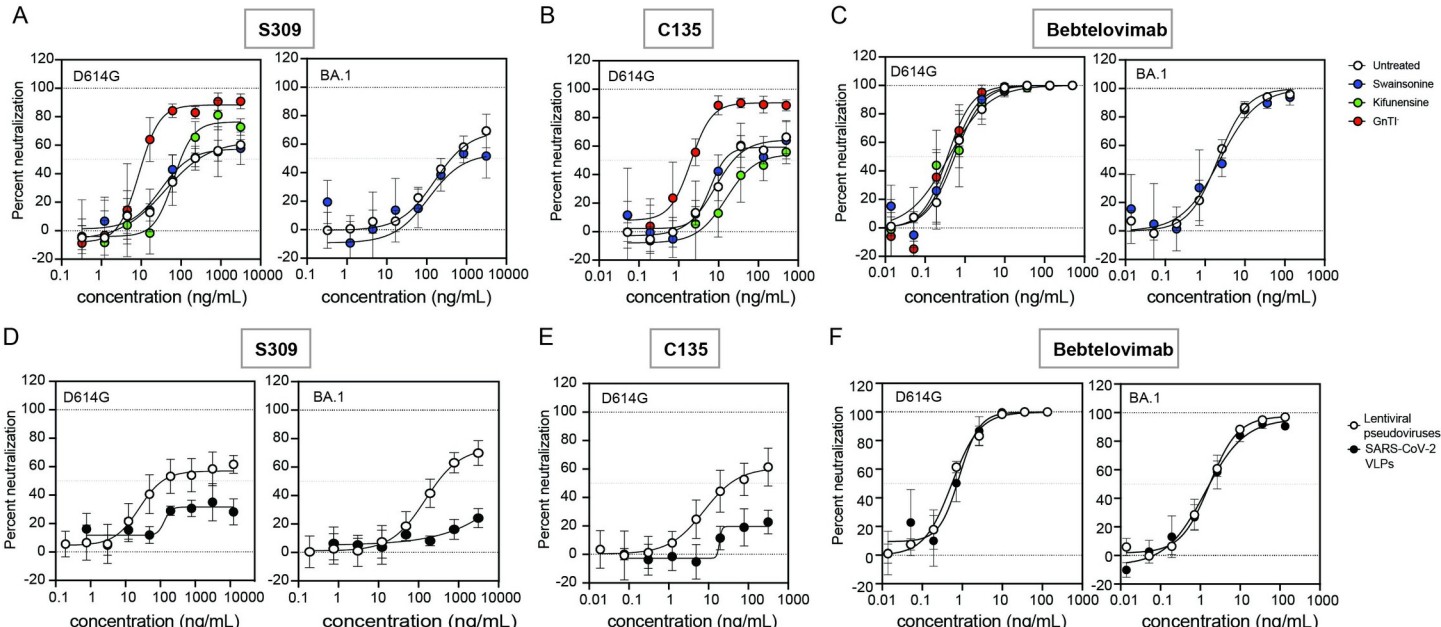

**Fig 6. Pseudovirus platform affects the maximum neutralization by class 3 antibodies.** Neutralization curves by (A) S309, (B) C135 and (C) Bebtelovimab of D614G or BA.1 pseudoviruses produced in 293T cells (white circles), GnTI⁻ cells (red circles), 239T cells with 20 μM swainsonine (blue circles) or 50 μM kifunensine (green circles). Neutralization curves by (D) S309, (E) C135 and (F) Bebtelovimab of D614G or BA.1 lentiviral pseudoviruses (white circles) or SARS-CoV-2 VLPs (black circles) produced in 293T cells. Experiments are averages from two independent experiments, each with an intra-assay duplicate.

fraction of the VLP preparation may have a glycan composition that hinders binding of the class 3 antibodies to spike. Because increasing the high-mannose content in the spike of lentiviral pseudoviruses expressed in GnTI⁻ cells or in the presence of kifunensine increased maximum neutralization, we speculate that SARS-CoV-2 VLPs may have a higher content of complex glycans at one or more specific positions.

## Conclusions

The extensive glycan coat on the SARS-CoV-2 spike serves important functions in modulating spike expression, stability, conformational dynamics, and escape from antibodies. Here we show that despite the conservation of the 22 glycan motifs in the spikes of SARS-CoV-2 variants, the effects of glycan deletions and modifications can differ significantly among the variants.

In contrast to the spikes of the earlier D614G and Delta variants, we show that the spikes of Omicron variants were inefficiently processed by furin and poorly incorporated into pseudoviruses when lacking glycans at N717 and N801. Omicron infectivity was also more sensitive to modifications in spike glycosylation than earlier variants. We further show that glycan site loss or glycan modifications can indirectly affect the potency of some neutralizing antibodies against the D614G but not the BA.1 variant. These glycan effects may be due to differences in how glycans affect conformational dynamics in different variant contexts, similar to what has been described for NTD and RBD glycans in the ancestral spike [53,54]. The highly mutated Omicron spike likely leads to glycosylation changes that help preserve spike function and stability. These results underscore the impact of cell substrates, spike expression platforms, and assay methods on glycan modifications that can affect assessments of spike function and virus neutralization. The findings also highlight how glycan interactions in spike variants may contribute to spike plasticity.

## Materials & methods

### Plasmids and cell lines

The gene to express full-length wild-type (D614G), Delta (B.1.617.2) and Omicron (BA.1, BA.2, BA.4 and BQ.1.1, Table 3) SARS-CoV-2 spikes or their N-to-Q or N-to-A substituted variants were codon optimized and cloned into pcDNA3.1(+) plasmid (GenScript, Piscataway, NJ). The HIV gag/pol (pCMVΔR8.2) and the Luciferase reporter (pHR'CMV-Luc) plasmids were obtained from the Vaccine Research Center (VRC, National Institutes of Health, Bethesda, MD). The CoV2-M-IRES-E (Addgene plasmid # 177938), Luc-T20 (Addgene plasmid # 177941) and CoV2-N-WT-Hu1 (Addgene plasmid # 177937) plasmids were a gift from Jennifer Doudna [59]. 293T-ACE2/TMPRSS2 cells stably expressing ACE2 and TMPRSS2 were established and maintained as previously described [58]. 293T (CRL-11268), GnTI⁻ cells (CRL-3022) and Vero E6 cell line (CRL-1586) were purchased from ATCC (ATCC, Manassas, VA). Cell lines were cultured in Dulbecco's modified essential medium (DMEM) supplemented with penicillin/streptomycin, non-essential amino acids, L-glutamine, HEPES and 10% fetal bovine serum at 37˚C with 5% $CO_2$. Swainsonine and kifunensine were purchased from Santa Cruz Biotechnology (Dallas, TX) and stocks were prepared in DMSO.

### Antibodies

Monoclonal antibodies S309, C135, and COV-2-2196 were made by transfecting $3.60 \times 10^8$ 293-Freestyle cells (ThermoFisher Scientific, Waltman, MA) in 240 mL FreeStyle 293 media (ThermoFisher Scientific) with 127 µg each of heavy and light chain expression plasmids for

**Table 3. Sequences of the SARS-CoV-2 spikes used in the current study.**

| SARS-CoV-2 Variants | Mutations in spike (compared to wild type Wuhan-Hu-1) |
|---|---|
| D614G | D614G |
| B.1.617.2 (Delta) | T19R, G142D, del156-157, R158G, L452R, T478K, D614G, P681R, and D950N |
| BA.1 (Omicron) | A67V, del69-70, T95I, del142-144, Y145D, del211, L212I, ins214EPE, G339D, S371L, S373P, S375F, K417N, N440K, G446S, S477N, T478K, E484A, Q493R, G496S, Q498R, N501Y, Y505H, T547K, D614G, H655Y, N679K, P681H, N764K, D796Y, N856K, Q954H, N969K, and L981F |
| BA.2 (Omicron) | T19I, del24-26, A27S, G142D, V213G, G339D, S371F, S373P, S375F, S376A, D405N, R408S, K417N, N440K, S477N, T478K, E484A, Q493R, Q498R, N501Y, Y505H, D614G, H655Y, N679K, P681H, N764K, D796Y, Q954H, N969K |
| BA.4 (Omicron) | T19I, L24-, P25-, P26-, A27S, H69-, V70-, V213G, G339D, S371F, S373P, S375F, T376A, D405N, R408S, K417N, N440K, L452R, S477N, T478K, E484A, F486V, Q498R, N501Y, Y505H, D614G, H655Y, N679K, P681H, N764K, D796Y, Q954H, N969K |
| BQ.1.1 (Omicron) | T19I, L24-, P25-, P26-, A27S, H69-, V70-, V213G, G339D, R346T, S371F, S373P, S375F, T376A, D405N, R408S, K417N, N440K, K444TL452R, N460K, S477N, T478K, E484A, F486V, Q498R, N501Y, Y505H, D614G, H655Y, N679K, P681H, N764K, D796Y, Q954H, N969K |

the respective antibodies (kindly provided by Peter Kwong, Vaccine Research Center, NIH) using 293fectin (ThermoFisher Scientific) transfection reagent. Cultures were incubated at 37˚C with 8% $CO_2$ at 125 rpm and harvested at 72 hours post transfection. Cells were removed by centrifugation and the supernatant was filtered with a 0.45 μm filter followed by a 0.20 μm filter. Antibody was purified from the supernatant using a HiTrap Protein-G column (Cytiva, Marlborough, MA). Purified IgG was buffer exchanged to DPBS (Corning, Corning, NY) and concentrated using an Amicon-15 Ultra centrifugal filter with 30 kDa molecular weight cut off (Millipore-Sigma, Burlington, MA) and the concentration was determined by OD280 with a conversion factor of 0.71 mg/mL per 1 OD280. Etesevimab and Bebtelovimab were generously provided by Eli Lilly (Indianapolis, IN).

## SARS-Cov-2 pseudovirus production and neutralization

HIV-based lentiviral pseudoviruses were produced as previously described [58]. Briefly, three plasmids, the spike pcDNA3.1(+) plasmid (4 μg), the pCMVΔR8.2 plasmid (HIV gag/pol, 5 μg) and the pHR'CMV-Luc plasmid (luciferase reporter, 5 μg) were co-transfected using Fugene 6 (Promega, Madison, WI) in 293T cells in 10% FBS/DMEM that was seeded the day before at 60–80% confluency in a 10 cm dish. For assessing the effect of glycosylation, the pseudoviruses were expressed as described above except for the following changes: plasmids were transfected in 293T cells in the presence of 20 μM swainsonine, 50 μM kifunensine or in GnTI⁻ cells. After 48 h, pseudoviruses were harvested and supernatant was filtered through a 0.45 μm low protein binding filters and aliquoted at -80˚C until further use.

Infectivity of each pseudovirus preparation was determined in 293T-ACE2/TMPRSS2 cells as previously described [58]. Briefly, 100 μL of pseudoviruses preparation was inoculated onto ACE2/TMPRSS2 plated at $3.0 \times 10^4$ cells/well in a 96 well dish. The infectivity was determined by measuring luciferase activity 48 h later (luciferase assay system, Promega, Madison, WI) and normalized for the HIV-1 p24$^{CA}$ in the inoculum. Results used at least two preparations of each pseudovirus and included at least two independent experiments.

The neutralization assays were performed using 293T-ACE2/TMPRSS2 cells as previously described [58]. Briefly, pseudoviruses with titers 0.8–1.5$^6$ relative luminescence units (RLU)/ mL were incubated with 4x serial dilutions of antibodies for 2 h at 37˚C. The virus-antibody

mixtures (100 μL) were used to inoculate plates with ACE2/TMPRSS2 plated the day before at $3.0 \times 10^4$ cells/well. The virus infectivity was determined 48 h later by quantifying the luciferase activity. Neutralization curves were normalized to virus only control. Neutralization curves were fitted using nonlinear regression curve [inhibitor] vs normalized response (GraphPad Prism, La Jolla, CA). The antibody concentration dilution corresponding to 50% neutralization was defined as $IC_{50}$. All experiments were done at least in duplicates each with an intra-assay duplicate. For those antibodies that reached <100% (partial) neutralization the neutralization curves were fitted using nonlinear regression curve [inhibitor] vs response, and the antibody concentration dilution corresponding to 50% neutralization of the maximum neutralization was defined as $pIC_{50}$.

## SARS-Cov-2 VLP production and neutralization

SARS-CoV-2 VLPs were produced by co-transfecting the following plasmids spike-pcDNA3.1 (+) (8 ng), CoV2-M-IRES-E (1.65 μg), Luc-T20 (5 μg), and CoV2-N-WT-Hu1(3.35 μg) using Fugene 6 (Promega, Madison, WI) in a 10 cm dish containing 293T cells in 10% FBS/DMEM seeded the day before at 60–80% confluency. VLPs were harvested at 48 h, and supernatants were then filtered through a 0.45 μm low protein binding filters, aliquoted, and stored at -80°C until further use. Infectivity of each VLP preparation was determined in 293T-ACE2/TMPRSS2 cells. Briefly, equal volume of pseudoviruses dilutions (100 μL) were inoculated onto 96 well plates with ACE2/TMPRSS2 plated the day before at $3.0 \times 10^4$ cells/well. The infectivity of the pseudovirus preparations were determined 48 h later (luciferase assay system, Promega, Madison, WI). The neutralization assays were preformed using 293T-ACE2/TMPRSS2 cells as previously described above. Results used at least two preparations of each pseudovirus and included at least two independent experiments.

## Live SARS-CoV-2 production and infectivity

To measure the effects of kifunensine on SARS-CoV-2 growth kinetics, kifunensine that was resuspended in DMSO was added to culture medium at a final concentration of 50 μM in 12-well plates containing confluent Vero E6 cells 1–2 hours prior to infection. Cells were infected at an MOI of ∼0.002 with 100 focus-forming units (FFU) of WA1/2020 (BEI, NR-52281), hCoV-19/USA/PHC658/2021 B.1.617.2 (BEI, NR-55611), or B.1.1.529 hCoV-19/USA/HI-CDC-4359259-001/2021 (BEI, NR-56475) at 37°C as described previously [61]. Supernatant samples were collected from each well for each virus and treatment combination (n = 4) and infectious virus concentrations were titrated using a fluorescent focus assay in H1299-hACE2 cells as described previously.

## Detection of spike glycoproteins in pseudoviruses

Pseudovirus expressing cells were lysed using 1 mL of 1% Nonidet P-40 (NP-40), 150 mM NaCl, 100 mM Tris (pH 8.0) buffer (lysis buffer). Cell lysates were clarified by centrifugation and diluted to 1x Laemmli buffer, 100 mM DTT and heated to 95°C for 10 min. Pseudoviruses (1 mL) were pelleted at 20,000 g for 2h at 4°C using at Tomy TX-160 ultracentrifuge. Pellets were resuspended with 100μL of 1x Laemmli buffer, 100 mM DTT and heated to 95°C for 10 min.

Denatured samples (20 μL) were loaded in a 4–20% gel and transferred to nitrocellulose membranes. The amount of S1, S2 and p24 proteins were quantified using rabbit polyclonal antibody against the RBD domain (Sino Biological; Cat: 40592-T62), 1A9 anti S2 mouse monoclonal antibody (GeneTex, Irvine, CA) and HIV-1 p24 hybridoma (183-H12-5C; Dr. Bruce Chesebro, AIDS Research and Reference Reagent Program, Division of AIDS, NIAID, NIH) respectively.

## Supporting information

**S1 Fig. Representative western blots of the pseudovirus lysates probed with S2 antibody.**
(A) D614G pseudoviruses. (B) BA.1 pseudoviruses. Anti-S2, and anti-p24 antibodies were used for detection.
(TIF)

**S2 Fig. Residues D796 and N801 undergo major conformational changes as the spike transitions from the pre-fusion to the post-fusion conformation.** The D796Y mutation in Omicron may affect glycan occupancy or processing at position N801. The S1 region is presented in semi-transparent cartoon representation. Glycans N709, N717, N801, N1098 and N1134 are presented as red sticks. Colored lines highlight the movement of residue D796 and glycans N801 from the pre-fusion to the post-fusion conformation. Omicron mutated residues are presented in green spheres. D796 and N801 are not resolved in the post-fusion conformation.
(TIF)

**S3 Fig. Infectivity of D614G, B.1.617.2, BA.1 and BA.2 VLPs.**
(TIF)

**S1 Data. Excel spreadsheet containing, in separate sheets, the underlying numerical data and statistical analysis for Fig panels 1C, 1D, 1E, 2B, 2D, 3B, 3C, 4A, 4B and 4C.**
(XLSX)

## Acknowledgments

We thank Lisa Parsons and Ira Berkower (U.S. Food and Drug Administration) for critical reading of the manuscript. We also thank the Eli Lilly, Peter Kwong (Vaccine Research Center, NIH) and the U.S. Department of Health and Human Services, Office of the Assistant Secretary for Preparedness as part of the U.S. Government COVID-19 response Therapeutics Research Team for providing reagents for this project.

## Author Contributions

**Conceptualization:** Sabrina Lusvarghi, Carol D. Weiss.

**Formal analysis:** Sabrina Lusvarghi, Carol D. Weiss.

**Funding acquisition:** Carol D. Weiss.

**Investigation:** Sabrina Lusvarghi, Charles B. Stauft, Russell Vassell, Brittany Williams, Haseebullah Baha.

**Methodology:** Sabrina Lusvarghi, Russell Vassell, Wei Wang, Sabari Nath Neerukonda.

**Project administration:** Sabrina Lusvarghi, Carol D. Weiss.

**Resources:** Tony Wang, Carol D. Weiss.

**Supervision:** Sabrina Lusvarghi, Carol D. Weiss.

**Visualization:** Sabrina Lusvarghi, Carol D. Weiss.

**Writing – original draft:** Sabrina Lusvarghi.

**Writing – review & editing:** Sabrina Lusvarghi, Charles B. Stauft, Russell Vassell, Brittany Williams, Haseebullah Baha, Wei Wang, Sabari Nath Neerukonda, Tony Wang, Carol D. Weiss.

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
