## [Decision Letter · Decision Letter 0]

13 Aug 2023

Dear Dr. Weiss,

Thank you very much for submitting your manuscript "D614G and Omicron SARS-CoV-2 variant spike proteins differ in the effects of N-glycan modifications on spike expression, virus infectivity, and neutralization by some therapeutic antibodies." for consideration at PLOS Pathogens. As with all papers reviewed by the journal, your manuscript was reviewed by members of the editorial board and by several independent reviewers. In light of the reviews (below this email), we would like to invite the resubmission of a significantly-revised version that takes into account the reviewers' comments.

The reviewers appreciated the significant body of work presented in this manuscript, and felt that these data could add significantly to the field, and be of broad interest. However, they raise several textual and experimental concerns that should be addressed prior to potential acceptance. A major concern is the heavy reliance on data from pseudotyped viruses with very limited experiments performed in live virus assays. The methodology is sometimes light on details making the quality of the data difficult to fully assess. The reviewers also suggest that more effort and thought go into linking these site-specific data to a potential model or mechanism for these findings, rather than a conclusion which which simply restates the data.

We cannot make any decision about publication until we have seen the revised manuscript and your response to the reviewers' comments. Your revised manuscript is also likely to be sent to reviewers for further evaluation.

Sincerely,

Penny L. Moore

Academic Editor

PLOS Pathogens

Guangxiang Luo

Section Editor

PLOS Pathogens

Kasturi Haldar

Editor-in-Chief

PLOS Pathogens

orcid.org/0000-0001-5065-158X

Michael Malim

Editor-in-Chief

PLOS Pathogens

orcid.org/0000-0002-7699-2064

The reviewers appreciated the significant body of work presented in this manuscript, and felt that these data could add significantly to the field, and be of broad interest. However, they raise several textual and experimental concerns that should be addressed prior to potential acceptance. A major concern is the heavy reliance on data from pseudotyped viruses with very limited experiments performed in live virus assays. The methodology is sometimes light on details making the quality of the data difficult to fully assess. The reviewers also suggest that more effort and thought go into linking these site-specific data to a potential model or mechanism for these findings, rather than a conclusion which which simply restates the data.

Reviewer's Responses to Questions

**Part I - Summary**

Reviewer #1: This study explores how glycan deletions/modifications affect Omicron BA.1 and D614G spike expression, processing, incorporation into pseudoviruses as well as virus infectivity and neutralization by antibodies. The main finding of this paper is the site-specific glycan deletions have a different effect on Omicron than for previous variants. Notably, the loss of glycosylation at residues 717 and 801 each conferred a loss of infectivity for Omicron but not for D614G or Delta variants, with the decreased infectivity correlated with decreased spike processing and incorporation into Omicron pseudoviruses. Further, the authors showed that complex glycans play a more important role in Omicron, with mannose-enriched Omicron pseudoviruses being non-infectious, whereas D614G or Delta pseudoviruses generated under similar conditions remained infectious. The authors also tested authentic SARS-CoV-2 grown with kifunensine which decreased titers more for BA.1.1 than for Delta or D614G. Some glycan deletions affect neutralization by antibodies at various regions differently in D614G vs. BA.1.

This study provide valuable information on spike biology about the role glycans play for different SARS-CoV-2 variants, particularly how their effect on the currently prevalent Omciron differs from previous variants. This is a technically sounds study and incorporates the higher throughput (and safer) pseudovirus neutralization assays with fewer but quality authentic coronavirus experiments with mutations of interest. The information about antibody binding with the glycan deletions will be useful for antibody development in the future.

Reviewer #2: A large number of studies have been performed in this paper, and they suggest that glycans at specific locations of Spike protein have a greater role in regulating omicron Spike function compared to D614G Spike. Omicron Spike is also more susceptible to inhibitors of glycosidases. Further, glycans may regulate viral susceptibility to neutralization. The findings are largely in line with previous published results, but they are valuable due to the differences noted related to the role of glycans in the original virus vs. the newer variants.

A main limitation of the study is that most of the work is focused on pseudotyped lentivirus which are processed differently compared to the actual virus. Thus, whether the functional modifications observed with lentivirus carry over to the authentic virus remains to be seen. This is particularly related to observations related to the incorporation of N717 and N801 Spike into the virion particles.

From a biomedical perspective, the most interesting result is related to the studies with kifunensine where the titer of the Omicron variant is dramatically reduced compared to D614G. For a paper to be published in PLOS Pathogens, I would recommend great effort to characterize this difference using additional assays related to CPE and plaque formation. A more careful investigation of this process can be of considerable basic science and translational value.

Reviewer #3: This manuscript represented a substantial body of work aimed at describing the impact of glycosylation on the spike protein of SARS-CoV-2 on viral infectivity, spike processing and neutralization sensitivity. The authors describe how despite high conservation of glycosylation sites between pre- and post-Omicron variants, modification of these glycan impact these variants uniquely. Removal of glycosylation sites in the S2, namely at N717 and N801, have profound impacts on Omicron variant pseudovirus infectivity, while only removal of N61 in S1 reduces infectivity of the D614G variant. The authors show that the presence of high mannose glycans ablates infectivity of Omicron variants, while having limited impact on pre-Omicron variants. Lastly the authors show that class 3 RBD mAbs are impacted by glycan modifications that favour the RBD up conformation of spike, while class 1 RBD mAbs are impacted by those modifications that favour the RBD down conformation.

What is lacking for me, is that the authors have not linked the findings from each set experiments to present a model or mechanism or reason for why, as their data suggests, there are functionally-dependent differences in glycosylation requirements for pre versus post-Omicron variants. The conclusion paragraphs that they authors have presented largely represents a summary of the results with implications that are very technical in their application.

**Part II – Major Issues: Key Experiments Required for Acceptance**

Reviewer #1: None noted.

Reviewer #2: I have suggested to strengthen Figure 4 as this is an important findings. Please see summary above.

Reviewer #3: Methods

Line 508

Given that the infectivity and relative infectivity measures are so critical to results, please provide more details as to how infectivity was measured and how relative infectivity was calculated. Is infectivity measured as the RLUs produced by a specific virus transfection? Were different aliquots or the same aliquot used for the p24 western blot and virus infectivity measures? Were multiple lots grown on different days compared?

VLP generation

Please include a section detailing how the SARS-CoV-2 VLPs were generated and quality checked.

Results

Line 436-444

This section comparing lentivirus to VLP is difficult to assess given the lack of details as to how the VLPs were generated. Regardless, results for a control antibody, that is able to neutralise the VLPs at greater than 30% maximum inhibition is required to demonstrate robustness of the assay.

Discussion

The authors have included multiple points in their results section that could form the basis of a well-constructed discussion, which is currently lacking and which I believe will force the authors to tie together the many different points into some mechanistic and biologically relevant implications, rather than simply a set of technical recommendations.

**Part III – Minor Issues: Editorial and Data Presentation Modifications**

Reviewer #1: It is unclear what the section on the VLPs is adding to this study.

Lines 330-333: “Both the apo and the S309 bound structures of the ancestral spike have an even proportion of closed and one-up RBD structures, but the one-up RBD conformation dominated the apo and S309-bound Omicron structures [48].” It is key to note here that the preference for one-up RBD in the Omicron spike is a result of the engineered 2P mutations, whereas without these mutations, the Omicron spike assumes a wider variety of conformational states (https://pubmed.ncbi.nlm.nih.gov/35447081/, https://pubmed.ncbi.nlm.nih.gov/35452593/). The authors should take this into account when discussing their data in the context of RBD up/down dynamics.

-In line 13, “Short title: Effects of glycosylation in Omicron and D614 spikes differ.” Should be changed to “Short title: Effects of glycosylation in Omicron and D614G spikes differ.”

-Lines 79 and 81: “The protein sequence determines the position of the N-linked glycosylation, but the glycan occupancy and composition depend on the three-dimensional protein structure and the accessibility to glycan-processing enzymes.” Citations should be added here.

-When discussing the Li et al. paper and Huang et al papers (lines 89-91), it may be useful to add information on the expression system used in these studies.

-Line 132: What the authors mean here is “deletion of the glycosylation site at N801”.

-Line 187: The acknowledgement that “differences in local structure may also affect glycan site occupancy or glycan modifications” is appreciated, especially because a residue change in Omicron BA.1 occurs so close to the glycan site. A figure indicating this structural proximity will be a useful addition here.

Reviewer #2: None

Reviewer #3: Title

I would recommend a modification to the title to read as “ Effects of N-glycan modifications on spike expression, virus infectivity and neutralization sensitivity in ancestral compared to Omicron SARS-CoV-2 variants”

Throughout the manuscript – “authentic virus”.

I would recommend that the authors refer to “authentic virus” as “live virus” or because in my opinion, there is nothing authentic about culturing SARS-CoV-2 in Vero E6 cells.

Line 52

Suggest to alter the sentence to read as follows “Changes in the overall glycan composition also reduced the infectivity of Omicron compared to D614G in the live or pseudovirus format.”

Line 67

The S2 subunit contains the transmembrane domain and contains other regions that do not span the transmembrane region. Therefore please modify the sentence to read “….and a non-covalently associated viral fusion subunit (S2).”

Line 119

Please clarify whether fold drop in infectivity was based on comparison to wild-type D614G (Fig 1C) or wild-type BA.1 (Figure 1D)?

Line 204

The authors state that D614G and BA.1 had similar levels of infectivity, however in the western blots in both Figure 2 and Figure 3, the S1 bands for BA.1 appear to be much darker than for D614G. Can the authors comment on this relative to the infectivity read out?

Line 227

The authors state that BA.1/BA/2 spikes depend on more “proper” glycosylation. Can the authors clarify what they mean by proper glycosylation or replace the term with “more processed glycans”?

Figures

Figure 1 legend

Please state what the dashed lines in panels C, D and E indicate?

Might make it easier for the reader if you indicate where the 10-fold and 100-fold difference in infectivity is for Fig 1, C, D and E.

Figure 3A

Please include labels for the various glycoforms eg high mannose/oligomannose, hybrid or complex. Alternately, because for eg Man5-9GlcNAc2 is referred to in the text, these types of labels may also help orient the reader.

Figure 3 legend

There appears to be an error in the description for panel B and C. I assume panel B to represent results for multiple variants infectivity on the absence of glycosylation modifiers, however the current description appears to refer to panel C. In addition, the description for panel C only refers to mean and duplicates for each assay.

PLOS authors have the option to publish the peer review history of their article (what does this mean?). If published, this will include your full peer review and any attached files.

Reviewer #1: No

Reviewer #2: No

Reviewer #3: **Yes: **Jinal N. Bhiman

Figure Files:

Data Requirements:

Please note that, as a condition of publication, PLOS' data policy requires that you make available all data used to draw the conclusions outlined in your manuscript. Data must be deposited in an appropriate repository, included within the body of the manuscript, or uploaded as supporting information. This includes all numerical values that were used to generate graphs, histograms etc.. For an example see here on PLOS Biology: http://www.plosbiology.org/article/info:doi%2F10.1371%2Fjournal.pbio.1001908#s5.
---

## [Decision Letter · Decision Letter 1]

30 Oct 2023

Dear Dr. Weiss,

We are pleased to inform you that your manuscript 'Effects of N-glycan modifications on spike expression, virus infectivity, and neutralization sensitivity in ancestral compared to Omicron SARS-CoV-2 variants' has been provisionally accepted for publication in PLOS Pathogens.

Before your manuscript can be formally accepted you will need to complete some formatting changes, which you will receive in a follow up email.  A member of our team will be in touch with a set of requests. At the same time, please address the remaining minor comments raised by Reviewer 3 below.

Best regards,

Penny L. Moore

Academic Editor

PLOS Pathogens

Guangxiang Luo

Section Editor

PLOS Pathogens

Kasturi Haldar

Editor-in-Chief

PLOS Pathogens

orcid.org/0000-0001-5065-158X

Michael Malim

Editor-in-Chief

PLOS Pathogens

orcid.org/0000-0002-7699-2064

Reviewer Comments (if any, and for reference):

Reviewer's Responses to Questions

**Part I - Summary**

Reviewer #2: The authors have addressed my questions, satisfactorily. This is an interesting paper.

Reviewer #3: As with the initial submission of this manuscript, the authors have generated a significant amount of data describing how glycosylation of the SARS-CoV-2 spike affects viral infectivity, spike processing and neutralization sensitivity. The inclusion of additional data/controls as well as details regarding the methodology has convinced me and addressed all my queries.

I like the description of the glycan belt and given that residues N801 and N796 are not resolved in the post-fusion spike structure, makes the role of these residues even more intriguing. Further, alterations in S2, including glycosylation, may affect overall conformation of S1, which in turn may affect for example receptor engagement. I find these data fascinating.

Lastly I thank the authors for addressing all comments and for being so generous in accepting my suggestions.

**Part II – Major Issues: Key Experiments Required for Acceptance**

Reviewer #2: (No Response)

Reviewer #3: None.

**Part III – Minor Issues: Editorial and Data Presentation Modifications**

Reviewer #2: (No Response)

Reviewer #3: Methods

Line 565

“The neutralization assays were preformed…” should be “The neutralization assays were performed…”

Figures

Figure 1 panel E – I would change the y-axis title to “Infectivity relative to parental variant”

Figure 3 panel C - I would change the y-axis title to “Infectivity relative to untreated variant”

Results

Line 238

“...appears to be more sensitive glycosylation modifications.” Should be “...appears to be more sensitive to glycosylation modifications.”

PLOS authors have the option to publish the peer review history of their article (what does this mean?). If published, this will include your full peer review and any attached files.

Reviewer #2: No

Reviewer #3: **Yes: **Jinal N. Bhiman

---

## [Editor Report · Acceptance letter]

5 Nov 2023

Dear Dr. Weiss,

We are delighted to inform you that your manuscript, "Effects of N-glycan modifications on spike expression, virus infectivity, and neutralization sensitivity in ancestral compared to Omicron SARS-CoV-2 variants," has been formally accepted for publication in PLOS Pathogens.

Best regards,

Kasturi Haldar

Editor-in-Chief

PLOS Pathogens

orcid.org/0000-0001-5065-158X

Michael Malim

Editor-in-Chief

PLOS Pathogens

orcid.org/0000-0002-7699-2064